# The Denominator Matters! Lessons from Large Database Research in Neonatology

**DOI:** 10.3390/children7110216

**Published:** 2020-11-07

**Authors:** Veeral N. Tolia, Reese H. Clark

**Affiliations:** 1Division of Neonatology, Baylor University Medical Center and Pediatrix Medical Group, Dallas, TX 75246, USA; 2The MEDNAX Center for Research, Education, Quality, and Safety, Sunrise, FL 33323, USA; reese_clark@mednax.com

**Keywords:** infants, infant, neonatal intensive care, health service research, statistics, study interpretation

## Abstract

Observational studies from large datasets are becoming more common in neonatology. In this review, we highlight the importance of the denominator in study design and interpretation including examples of bias from source data, weight-based categories, age-related bias, and diagnosis-based denominators.

## 1. Introduction

Observational studies from “big data” sources are important for generating hypotheses and for doing comparative effectiveness studies when randomized controlled trials are not feasible. In neonatology, we are fortunate to have several high-quality datasets, including from National Institute of Child Health and Development [1], the Canadian Neonatal Network [2], the Children’s Hospitals Neonatal Consortium [3], the Vermont Oxford Network (VON) [4], the California Perinatal Quality Care Collaborative [5], and our own dataset—the Pediatrix Clinical Data Warehouse (CDW) [6].

Clinically relevant insights have been reported from all of these sources, and these studies are becoming more common every year. Therefore, we must be intentional in the design and interpretation of findings from these sources. Critical review of these studies tends to focus primarily on the observational numerator—such as how the exposure or outcome is defined or how to account for confounding factors. We feel that the denominator deserves similar scrutiny. 

Our goal with this review is to demonstrate how the denominator can be a hidden source of bias in retrospective observational research. A better understanding of these issues can help both clinicians and researchers in applying appropriate context to this type of data. 

## 2. Ascertainment Bias in the Source Population

Ascertainment bias refers to bias that results from the sampling method. All of the neonatal databases have specific enrollment criteria for an infant to be included in the dataset. Notably, these criteria are unique to each dataset, and these differences can have important ramifications on study design. 

In 2015, Rysavy et al. [1] published an insightful observation about mortality at periviable gestations. He found that “differences in practices regarding the initiation of active treatment in extremely preterm infants appear to explain a large portion of the between-hospital variation in survival among such patients”. Although we would have liked to validate this observation across our ~300 centers, this study would not have been possible using the Pediatrix CDW, which only collects data on infants that are admitted to the neonatal intensive care unit (NICU). None of the infants who died in the delivery room could be included in the denominator in a CDW study examining the same relationship. 

This issue extends to the other end of the gestational age spectrum as well. The criteria for inclusion of larger and more mature infants in many of these datasets select for infants who are also ill enough to require critical or intensive care. This is a minority of term infants (Figure 1), and so, studies evaluating less severe diseases in term infants (such as neonatal hypoglycemia or hyperbilirubinemia) sourced from the CDW (or other NICU-based datasets) are likely to under-report disease incidence. 

One method to account for this is to include an unrelated but similar diagnosis that can act as an internal control. When we reviewed the changing prevalence of gastroschisis, we used the prevalence of patients with omphalocele to act as an internal control. We assumed two similar gastrointestinal anomalies would vary in a similar way if referral or care parents explained the changes. We found that the prevalence of gastroschisis changed in significant ways but the prevalence of omphalocele was much more consistent [6].

Gestational age is not the only cause of bias in source data. Cohorts derived from Children’s Hospital NICUs have a referral bias in that the infants admitted to their sites are more likely to have complex diagnoses or require specialized care [3]. The effect of referral bias goes both ways. A level 2 NICU may have a zero mortality rate if that NICU consistently refers all of its critically ill infants to a regional level 4 NICU. Transferred patients are sometimes included in the denominator, but because their final outcome is not known, their data are not included in the numerator. 

Studies of infants with a poor prognosis (such as hydrops fetalis or genetic anomalies) can have ascertainment bias from several sources: prenatal diagnosis may result in pregnancy termination, and infants who are receiving palliative care may receive care in the NICU. These factors complicate comparisons between datasets and can limit overall study generalizability, so it is essential to understand the context of source data. 

## 3. Selection Bias from Weight-Based Cohort Selection 

In neonatology, it is common to define infant categories based on birth weight. Study cohorts will select for very low birth weight (VLBWs) or extremely low birth weight infants (ELBWs). Although these definitions are based on objective measures that are accurate, reliable, and easily reproducible, we find this classification a problematic source of bias for several reasons. 

First, VLBW infants represent an extremely heterogeneous group (Figure 2). The mortality rate varies from 47% in the infants <500 g to <1.9% in the infants who weigh 1251 to 1499 g at birth. Mortality is obviously a principal outcome, but even when it is not the primary outcome, its role as a competing outcome or in immortal time bias can greatly influence study validity in a group of infants with such heterogeneity. 

Second, the gestational age at birth VLBW infants is not normally distributed. Because of this, more mature and larger infants are over represented in VLBW cohorts. For example, in a VLBW cohort from the Pediatrix CDW, almost half of infants are greater than 29 weeks and the ratio of 28 week infants to 23 infants is 3:1 (Table 1). One method used to mitigate this effect is to bracket VLBWs by gestational age; VON allows reports to be restricted to ≥23 weeks and ≤29 weeks. However, applying that filter to the CDW would exclude 35% of the sample, and there would be no impact on the 3:1 ratio of infants at 28 weeks to 23 weeks. The ELBW classification is also problematic, in that it also skews the data but in a different way. Because of changes in the relative distribution of gestational ages, an ELBW cohort actually includes more 23 week infants than 28 week infants. 

A third concern with weight-based classification for premature infants is bias introduced by small-for-gestational-age infants. Older studies suggested that growth restriction, presumably caused by some process that accelerates fetal maturity, may actually improve some morbidities such as respiratory distress syndrome [7]. However, within each birth weight group, infants with growth restriction were significantly more advanced in gestational age, potentially giving rise to the impression that these infants do better than expected for their birth weight as opposed to their gestational ages. More recent work was verified that a prenatal history of intrauterine growth restriction (IUGR) and being born small for gestational age (SGA) are associated with an increased risk of mortality and morbidity and poor long term outcomes both in term and preterm infants [8,9,10].

For these reasons, we prefer to use gestational-age-based categories (such as last completed week or the extremely low gestational age newborns or ELGANs [11]), despite the intrinsic imprecision in measurement of gestational age. 

## 4. How Age Influences the Denominator

Complications can also arise in any denominator that is age-dependent due to skews in distribution of NICU stay and/or timing of death. 

The characteristics of the study cohort included in an age-based study cohort changes dramatically with duration of NICU stay. For example, the incidence of early onset sepsis is dependent on the denominator. Most blood cultures are done in the first 3 days after birth (Figure 3a), and that large denominator, which includes term and late preterm infants at low risk for having a positive blood culture means that the incidence of a positive culture is low early in the hospital course (Figure 3b). After 3–5 days, these larger, healthier, lower-risk infants are discharged and the infants that remain in the hospital are less mature, sicker, and at higher risk of having a positive blood culture related to their illness, environmental exposures, and invasive procedures. The numerator/likelihood of a positive culture is also changing. The measured incidence of early onset sepsis will be higher if the cohort of infants includes infants in the hospital for ≤7 days instead of limiting the study cohort to include only infants who were discharged before ≤3 days of age. 

Mortality poses a similar problem. In premature infants, mortality rates are the highest in the first few days after admission and every day that these babies survive, they are more likely to go home alive [12]. As a result, cohorts that are derived from older premature infants (which seems reasonable when the exposure of interest occurs late in the hospital stay) will have a survival bias compared to a cohort that includes all premature infants. This effect may help explain the wide variation in rates of retinopathy of prematurity described in a recent publication that compared international experiences of the disease [13].

## 5. Beware the Dynamic Denominator 

Sometimes, in order to further explore disease incidence of progression, investigators will define their cohort as infant with a specific diagnosis. However, bias can be introduced by changes in the evaluation or classification of the disease of interest. This leads to a denominator that changes over time or among studied groups.

We recently described changes in the frequency of patent ductus arteriosus (PDA) diagnosis in premature infants [14]. In that same paper, we also described changes in PDA treatment patterns in all NICU infants. Because of the underlying changes in PDA diagnosis, a study that evaluated treatment changes over time with a denominator limited to infants with a PDA would have led to different results. We illustrate this in Figure 4, which shows changes in PDA treatment over time in two different denominators: all infants or those infants diagnosed with a PDA. Although the general trend is similar, the decrease in treatment among the cohort of all infants was 23% compared to a 28% decrease among infants diagnosed with a PDA. Notably, the relative difference in treatment rates in these two cohorts varied over time as well, from 17% in 2010 to 11% in 2019. 

Another example comes from disease incidence of intraventricular hemorrhage (IVH) in preterm infants. There is substantial variation in diagnostic cranial ultrasound (CUS) in higher gestational age groups [15] (29 weeks through 32 weeks), and some authors have suggested a risk-based screening approach [2]. We sought to understand how the rate of screening CUS might influence the incidence of IVH, so we calculated the rate of screening of these infants by center and then stratified centers into three groups based on each center’s rate of CUS in this population. We found that the disease incidence of IVH was higher in the centers with greater rates of CUS (Table 2). 

As these two cases illustrate, changes in disease incidence or measurement must be considered potential sources of bias when the denominator is based on a diagnosis. 

## 6. Summary

Big data research is important. The large sample sizes are almost always able to discern statistically significant relationships. Randomized trials are not available or feasible for many pressing clinical questions in our field. These examples come from the Pediatrix Clinical Data Warehouse [16]. The source of CDW data is medical records from approximately 350 NICUs that are managed by MEDNAX, Inc. (Sunrise, FL, USA)—approximately one fourth of NICU admissions in the United States. Despite its size, the CDW has several limitations. It is not geographically representative. The data are generated from physicians’ documentation, and some information might be better obtained via a standardized case report form (a method used by the Vermont Oxford Network). Similarly, each neonatal dataset has its own set of unique limitations [17]. There are addition limitations to all United States NICU data that is currently collected [18]. How one defines the denominator when using these sources can introduce bias and influence the study results, validity, and generalizability. For this reason, we urge everyone to think critically about both the numerator and denominator—they both matter. 

## Figures and Tables

**Figure 1 children-07-00216-f001:**
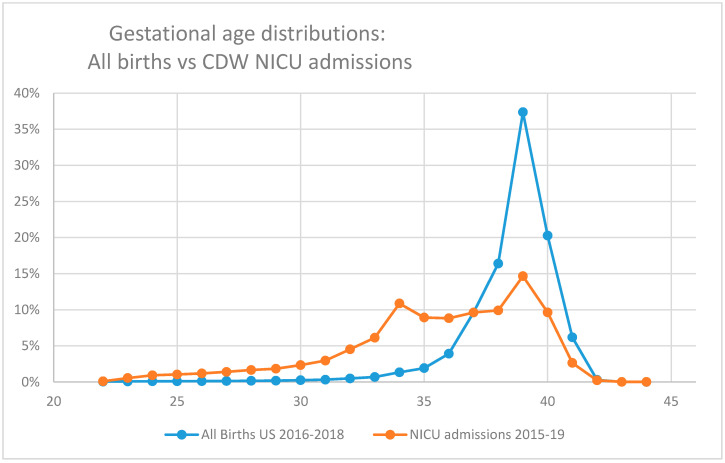
Gestational age distributions: all births vs. neonatal intensive care unit (NICU) admissions. Data derived from the Centers for Disease Control and Prevention and the Pediatrix Clinical Data Warehouse (CDW).

**Figure 2 children-07-00216-f002:**
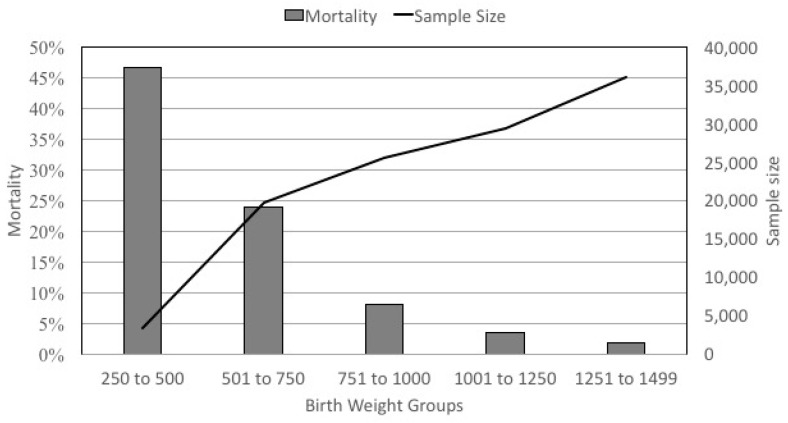
Mortality rate and sample size in very low birth weight infants stratified by birth weight in the Clinical Data Warehouse (CDW) from 1997–2019.

**Figure 3 children-07-00216-f003:**
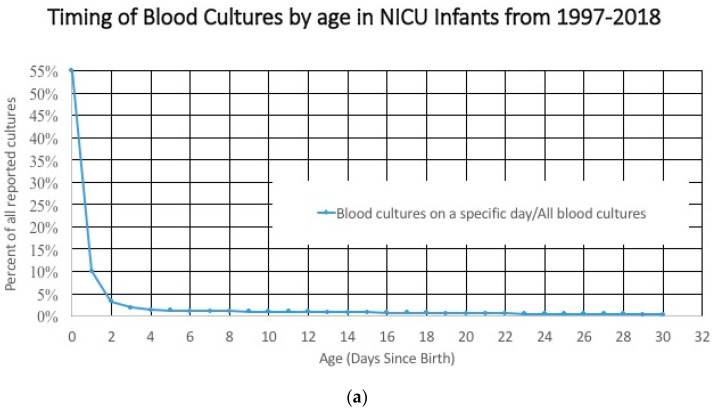
(**a**) Timing of blood cultures by age in neonatal intensive care unit (NICU) infants. (**b**) Frequency of a positive blood culture by age in NICU infants.

**Figure 4 children-07-00216-f004:**
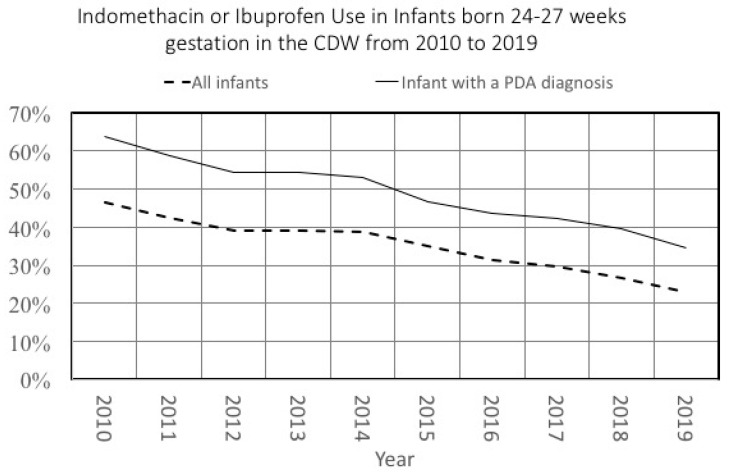
Treatment with Indomethacin or Ibuprofen in infants diagnosed with a patent ductus arteriosus (PDA) or in all infants 24–27 weeks.

**Table 1 children-07-00216-t001:** The distribution of gestational ages in infants in the CDW from 2008–2018.

	All VLBW Infants		VLBW Infants ≥ 23 and ≤ 29 Weeks		All ELBW Infants	
GestAge	N	%	N	%	N	%
22	879	0.77%	0	0.00%	879	1.81%
23	4855	4.26%	4855	6.54%	4853	10.00%
24	8320	7.30%	8320	11.21%	8308	17.12%
25	9264	8.13%	9264	12.49%	9066	18.68%
26	10,589	9.29%	10,589	14.27%	8915	18.37%
27	12,407	10.89%	12,407	16.72%	6856	14.13%
28	14,465	12.69%	14,465	19.50%	4561	9.40%
29	14,294	12.54%	14,294	19.27%	2429	5.00%
30	13,238	11.62%	0	0.00%	1391	2.87%
31	9880	8.67%	0	0.00%	681	1.40%
32	7548	6.62%	0	0.00%	379	0.78%
33	3867	3.39%	0	0.00%	129	0.27%
34	2595	2.28%	0	0.00%	54	0.11%
35	959	0.84%	0	0.00%	18	0.04%
36	454	0.40%	0	0.00%	13	0.03%
37	176	0.15%	0	0.00%		
38	73	0.06%	0	0.00%		
39	48	0.04%	0	0.00%		
40	30	0.03%	0	0.00%		
41	13	0.01%	0	0.00%		
42	8	0.01%	0	0.00%		
43	1	0.00%	0	0.00%		
44	5	0.00%	0	0.00%		
All	113,968	100.00%	74,194	100.00%	48,532	100.00%

**Table 2 children-07-00216-t002:** Rates of cranial ultrasound screening and intraventricular hemorrhage in infants 29–32 weeks from infants in the CDW from 2008–2017.

Centers Stratified by Screening Rate (Number of Infants)	Screening Rate	Rate of Any IVH
29 Lowest Third (*n* = 1449)	63%	4.97%
29 Middle Third (*n* = 11185)	92%	7.66%
29 Highest Third (*n* = 4185)	94%	8.84%
30 Lowest Third (*n* = 1789)	47%	3.24%
30 Middle Third (*n* = 14412)	87%	5.16%
30 Highest Third (*n* = 5279)	93%	6.14%
31 Lowest Third (*n* = 2489)	35%	1.93%
31 Middle Third (*n* = 17945)	76%	3.19%
31 Highest Third (*n* = 6596)	90%	3.87%
32 Lowest Third (*n* = 3907)	14%	0.64%
32 Middle Third (*n* = 27220)	47%	1.59%
32 Highest Third (*n* = 9906)	78%	2.28%

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
