# Peer review of "The Denominator Matters! Lessons from Large Database Research in Neonatology"

_children, 2020, doi:10.3390/children7110216_

Round 1

Reviewer 1 Report

Thank you very much for submitting this review to "Children".

This work is an interesting and timely warning about important biases in large database studies that too often go unnoticed by researchers. In general, it is very well written and is very clarifying of the problem to which it refers.

Just a few small remarks for your consideration:

  1. Please, double-check the reference format to follow the journal's guidelines.
  2. Consider correcting some possible "typos":

    Line 28: … such at how a the exposure… such as how the exposure (?).

    Line 66: …because their their final outcome is not known, … 

    Line 74: … extremely low birth weight infants (ELBWs). extremely low birth weight (ELBWs) infants. 

    Line 80: … can greatly effect study validity… affect (?).

    Line 127: … first few days after admission and that (?) every day that these babies survive, …

    Line 131: … variation in rates of retinopathy of prematurity rates (?) described…

    Line 138: We recently reported on changed (changes?) in the frequency of patent ductus arteriosus…

Author Response

Thank you for your thorough review. We apologize for the typographical errors. We have corrected the errors that you identified and have also made a few additional corrections. 

Reviewer 2 Report

The report provides clear illustrations of the importance of denominator bias and implies an approach for assessing the presence of denominator bias for the reader/reviewer of studies or for the researcher who is formulating his/her study's design. Though it is not a comprehensive consideration of the topic it covers some of the most important issues in denominator bias. I think it provides useful material for educational/training sessions concerning study design. 

The examples provides in the report are all from the Pediatrix CDW. Though this is implied in the title of the report I think it would be helpful for the authors to underscore this in the body of the report, too. It would helpful to include a comment on the ways in which the CDW dataset is representative and unique among other large datasets. There are some publications that could be referenced to address this point: here are a few

Spitzer, A.R., D. Ellsbury, and R.H. Clark, The Pediatrix BabySteps® Data Warehouse — a unique national resource for Improving outcomes for neonates. The Indian Journal of Pediatrics, 2015. 82(1): p. 71-79.

Goodman, D.C. and G.A. Little, Data deficiency in an era of expanding neonatal intensive care unit care. JAMA Pediatrics, 2018. 172(1): p. 11-12.

Statnikov, Y., B. Ibrahim, and N. Modi, A systematic review of administrative and clinical databases of infants admitted to neonatal units. Archives of Disease in Childhood - Fetal and Neonatal Edition, 2017.102(3): p. F270-F276.)

Table 2 has a number of tercile groupings but it is unclear to me how the groups differ: it would be helpful to clarify how those groups differ.

Author Response

Thank you for your careful review of our manuscript. We have  expanded the summary section to provide some additional information about the Pediatrix CDW and included the recommended citations. We have also rewritten the description of cohort delineation for the data presented in table 2.